# Food Consumption and Emotions at a Salad Lunch Buffet in a Multisensory Environment

**DOI:** 10.3390/foods9101349

**Published:** 2020-09-23

**Authors:** Ulla Hoppu, Sari Puputti, Saila Mattila, Marjaana Puurtinen, Mari Sandell

**Affiliations:** 1Functional Foods Forum, University of Turku, 20014 Turku, Finland; ulla.hoppu@utu.fi (U.H.); sari.puputti@utu.fi (S.P.); saila.mattila@utu.fi (S.M.); 2Department of Teacher Education, University of Turku, 20014 Turku, Finland; marjaana.puurtinen@utu.fi; 3Department of Food and Nutrition, University of Helsinki, 00014 Helsinki, Finland

**Keywords:** lunch buffet, vegetables, food intake, multisensory, emotion terms

## Abstract

The food experience is multisensory and multisensory external stimuli may affect food choice and emotions. The objective of this study was to evaluate the effect of a multisensory eating environment on food choice, intake and the emotional states of the subjects in a salad lunch buffet setting. A total of 30 female subjects consumed a salad lunch twice in the multisensory laboratory. The two test conditions (control and multisensory condition with environmental stimuli) were randomized and the visits were scheduled one week apart. Subjects selected and ate a meal from a salad buffet including 14 food items and the intake of each item was weighed. They answered an online questionnaire about the meal and their emotional states (20 different emotion terms) after the lunch. There was no significant difference in the food consumption between the control and multisensory conditions. The subjects were very satisfied with their lunch for both study visits but the pleasantness of the eating environment was rated higher under the multisensory condition. In emotional terms, the subjects selected the term “happy” significantly more frequently under the multisensory condition compared with the control. In conclusion, the multisensory eating environment in this study was not related to food intake but may be associated with positive emotions. The effect of the eating environment on food choice and experience deserves further study with a larger study population in a real lunch restaurant setting.

## 1. Introduction

The workday lunch is an important part of Finnish food culture [1] and lunchtime salad buffets are common at restaurants and worksite canteens in Finland. Lunch is typically eaten around noon in Finland and on workdays the lunch break is half an hour. Healthy lunch choices, especially consumption of vegetables may promote public health [2] and, when replacing less sustainably produced food items in diet, also sustainable food system. Recently, different nudging techniques have been used to increase vegetable intake [3]. Lunch breaks spent in pleasant environments may be associated with positive emotions and further with wellbeing and recovery from stress [4]. Customers value peaceful eating environments but they may have different expectations for interior colors, background music and desired emotional sensations in relation to the restaurant menus [5].

Food perception is multisensory, integrating taste, smell, vision, touch and hearing. Food items have various internal sensory attributes. For example, the perception of vegetable quality combines many sensory characteristics [6,7]. External sensory stimuli, such as visual or sound, in the eating environment may modulate the multisensory experience [8]. The eating context, for example at home, a lunch restaurant or snack bar, may provide various external stimuli affecting food choices and perception. Traditionally, sensory evaluations have been performed in standard sensory laboratory conditions. Recently, different multisensory, immersive or virtual reality applications have been tested in consumer research [9]. Most previous studies involved the evaluation of single foods or beverages in virtual or multisensory conditions such as cookies [10], coffee [11], beer [12] or non-alcoholic beer [13]. Zandstra et al. reported a consumer study of tomato soup comparing tasting sessions in three different contexts: in a laboratory, an immersive simulated café and a real café [14]. A virtual reality eating environment was used in consumer studies evaluating snack products and emotions [15] as well as chocolate products and emotions [16]. To our knowledge, food consumption at a salad buffet in a multisensory environment has not been studied previously.

The emotions elicited by different food products have also been a research focus recently but the entire eating situation has seldom been evaluated using emotional terms. Various methods have been used to evaluate emotions evoked by food experiences including questionnaires with emotion, mood or wellness terms [17,18]. Recently, new types of methods such as a language-independent graphical tool with emoji have been developed for the assessment of food-elicited emotions [19]. Emotion questionnaires have been used in addition to sensory tests to identify the differences between tested products and even to predict food choice [20,21]. Different eating environments have been associated with different emotions [22] and thus may be related to the consumer’s experience of the meal.

The main aim of the present study was to evaluate the effect of a multisensory eating environment on food intake, especially vegetable and fruit intake, in a salad lunch buffet setting. A further aim was to compare subjects’ reported emotional states under control and multisensory conditions.

## 2. Methods

### 2.1. Subjects

The subjects had previously participated in extensive sensory tests [23] and further taste testing in a multisensory laboratory. Invitations to participate in this study part were sent to 62 female subjects. Two sets of data were collected in this study: food consumption and emotions as well as eye-tracking data, which were used to record the lunch sessions in this study and are reported in more detail in another article (submitted manuscript). Celiacs and pregnant or breastfeeding women were excluded and subjects with smell hypersensitivity were not recommended to participate. Due to eye-tracking data collection, normal vision was required (below −1.0 diopter). Wearing contact lenses was allowed but wearing glasses was not permitted during food selection and eating. Wearing reading glasses was allowed when answering the online questionnaire. Food allergies and intolerances of the subjects were enquired about before the study visit and just before the meal. A total of 32 subjects attended the first visit but one did not attend for the second and one subject was excluded due to noncompliance with the study protocol (having lunch elsewhere and taking only a small portion of salad). Thus, 30 subjects attended the required two sessions and provided complete data for analysis. The study protocol was approved by the Ethical Committee of the University of Turku and all subjects provided written informed consent. The lunch was free for the subjects and no other compensation for participation was offered.

### 2.2. Buffet Foods

The lunch buffet included 14 different food items. The foods, their preparation and serving sizes are described in Table 1. Food items were ordinary foods generally included in lunch salad buffets in Finland. Food was selected based on visual appearance with mainly color pairs (red, green, orange, black, white, beige) so that they formed a colorful buffet. Foods also had different dominant taste qualities (salty, sweet, sour, bitter). Fresh vegetables and fruits were the main options. Two different lactose-free cheeses, chickpeas and peanuts were provided as protein sources. Pasta with two different sauces (pesto or aioli mayonnaise) was served to supply energy (carbohydrate and fat) for the lunch.

Food items were delivered weekly by the same local supermarket and the quality of the vegetables and fruit was carefully monitored daily. The food was prepared fresh daily in the kitchen of the sensory laboratory just before the session for each participant. After finishing the preparation, the serving trolley was kept in the cold storage room at +8 °C. The weights of the served and consumed amount of the foods were measured with a scale (Mettler Toledo PB3002-S, Mettler Toledo International Inc., Columbus, OH, USA), to 1 g accuracy. The foods were served in square-shaped 15 × 15 cm glass bowls. The bowls were placed on the serving trolley on three different levels (Figure 1). The order of the serving bowls was randomized for every subject.

The serving sizes were selected based on similar volume appearance in the bowls and so that the subjects felt that they could take enough. Serving tools were ordinary tablespoons, except using a salad server for lettuce. The plates were white porcelain with a diameter of 22 cm. In addition to the salad buffet, rye and oat bread as well as margarine were served. Olive oil with a lemon flavor and French dressing were also offered. Water was served as a drink with the meal and coffee or tea with biscuits were offered after the meal.

### 2.3. Multisensory Laboratory Conditions

Multisensory conditions with different landscapes, sounds and odors were pilot-tested beforehand. Of the pilot-tested options, the forest landscape with birdsong and orange scent were selected for the multisensory condition and the other condition was a plain control. These two different conditions (control vs. multisensory) were randomized between the first and second study visit for each subject. Thus, all participants attended in both conditions but in a randomized order.

The multisensory laboratory equipment included an odor diffuser (Pump unit BB-200, @aroma GmbH, Berlin, Germany) and controlled illumination (five bulbs on the wall and three bulbs in a floor lamp, Hue, Philips, Amsterdam, Netherlands). The audio-visual multimedia system included an 80-inch Apple-tv (Apple Inc., Cupertino, CA, USA) and an audio system with two speakers (Genelec Oy, Iisalmi, Finland). In the control condition, the neutral room lighting system was used. In addition, there was no sound, no scent and no visual landscape on the screen. In the multisensory condition, there was a landscape of a pine forest and lake during summertime and bright lighting, matching the color tones of the landscape on the screen (Figure 2). The soundscape was birdsong in a Finnish summertime forest with various species of bird (recorded in Kortesjärvi in June). Orange scent (Orange Oil Sweet Brazil Pera, @aroma GmbH, Berlin, Germany) was diffused to the room with an odor diffuser for 30 s before a subject entered the room and then for 5 s in 3 min intervals until the subject had finished eating.

### 2.4. Questionnaire

Subjects also answered an online questionnaire (Webropol Oy, Helsinki, Finland) on an iPad (Apple Inc., Cupertino, CA, USA). Before the lunch, the questions asked about how hungry they felt (four options from not at all to very hungry) and the time (hours and minutes) since their previous meal. After the meal, the questions included how full they felt (four options, not at all to very full) and if they were satisfied with the salad meal (four options, not at all satisfied to very satisfied). The liking of the test environment in the multisensory room was also evaluated (9-point scale). The emotion terms were selected and modified from emotion questionnaires [24,25]. The term selection was pretested with Finnish consumers in a previous study setting focusing on ambient odors in the multisensory room. Altogether, 19 emotion terms, both positive and negative, as well as an open question option (something else) were presented and the subjects could choose as many options as they liked (check all that apply) based on how they felt at that moment. After the second study session, a few background questions (education, weight, height, how often they have salad for lunch) were asked.

### 2.5. Procedure

The test sessions were organized at the multisensory laboratory of the Functional Foods Forum (University of Turku, Finland) at usual lunch times in Finland. Subjects were asked to attend two study sessions at the same time of day (either at 10:45 a.m. or 12:30 p.m.) at least one week apart. The sessions lasted approximately 30–45 min. Session conditions (control vs. multisensory) in the multisensory room occurred in randomized order. Subjects were also instructed to have the same kind of breakfast on both study days and they were asked to avoid the use of scented cosmetic products before visits.

Subjects were first asked to view the trolley for 20 s while the researcher stood next to her. Next, the researcher left the room and closed the door and the subject collected a meal from the buffet. The subjects were instructed to take as much as they wanted and have all the foods at once. The researcher then removed the trolley from the room. Subjects were seated alone in the multisensory room, ate their meal at their own pace and knocked on the door when they had finished eating. After the meal they were served coffee or tea with biscuits. Subjects answered the online questionnaire while having coffee or tea.

The test session was recorded with a head-mounted eye-tracker (Tobii Pro Glasses 2, Tobii AB, Danderyd, Sweden). The subject wore the wireless eye-tracking glasses and gaze data were sent to a laptop in another room in live video format. This allowed the researcher to monitor the session behind a closed door and no other video cameras were needed. The subject knew that she was being monitored and that she herself was not visible on the video since it only recorded her first-person view of the laboratory. The eye-tracker was removed when the subject began to answer the online questionnaire. For this study, the eye-tracking recording was used for monitoring the session and calculating the time spent on eating the self-selected salad. The eating time was measured from the time the participant sat down to eat until she knocked on the door and let the researcher know she had finished. The detailed description of the eye-tracking methodology and analyses are reported in another article; this paper focuses on the food intake and emotional measures.

### 2.6. Statistics

The basic results of the intake of foods are presented as means (SD). The same subject attended under both conditions and the intakes in control vs. multisensory conditions were compared using the Wilcoxon signed-rank non-parametric test for repeated measurements. For paired nominal data, McNemar’s test was applied. The statistical software used was IBM SPSS Statistics 26 (IBM Corporation, Armonk, NY, USA).

## 3. Results

The mean age of the participants (*n* = 30) was 53 years (SD 14 years) and their mean BMI was 26.8 kg/m^2^ (SD 6.9). The educational background was high; 50% had a university education and 30% had a university education with an applied sciences degree. Half of the subjects (50%) reported having salad for lunch one to three times per week and 27% one to three times per month. There was no difference in the time since the previous meal for the two study visits: control of 3.9 (SD 2.1) vs. multisensory of 4.0 (SD 2.8) hours. The state of hunger did not differ significantly either as 77% in the control and 73% in the multisensory condition felt very or fairly hungry before the lunch.

The number of food items taken altogether from the salad bowls varied; the range was 7 to 14, mean 11 (SD 1.5) but we found no significant differences between the control and multisensory conditions. The mean (SD) weights of foods and the total portion weight in the control and multisensory conditions are presented in Table 2. The sum variables of the food groups (vegetables, fruit, cheese and pasta) were calculated and analyzed but the intakes of these groups did not differ between the two conditions. Because the total weight of the portions differed between subjects, proportions (%) of the foods in the total portion weight were calculated. However, no significant differences in these variables were observed between study conditions.

The eating time did not differ significantly between the two conditions (control 12.8 min vs. multisensory 13.0 min). Considering fullness after the meal, the proportions of responses were exactly the same for both conditions: 47% reported feeling very full and 50% fairly full. Contentment with the salad was also good for both conditions. In the multisensory condition, 83% were very satisfied with the salad compared with 77% in the control condition (non-significant difference). Liking ratings of the testing environment differed (Figure 3). Overall, the multisensory condition was significantly more pleasant than the control (*p* < 0.001).

The selected emotion terms under the control and multisensory conditions are presented in Figure 4. Most of the selected terms were positive for both conditions. Over two-thirds of the participants felt healthy in both test environments. No one stated that they felt stressed, cold or tired in either condition. Respondents chose the term “happy” more often in the multisensory condition (*n* = 13) compared with the control (*n* = 5); *p* = 0.02, McNemar’s test. The subjects also tended to select the terms “relaxed” (*p* = 0.09) and “strong” (*p* = 0.07) more often for the multisensory condition.

## 4. Discussion

In the present study, no general effect of the multisensory environment on food choice or intake at a salad lunch buffet was observed. Therefore, changing individual food preferences and consumption patterns simply with external multisensory stimuli appears to be challenging. The selected foods, portion sizes and eating times were surprisingly similar for the same person under both conditions. This finding may reflect the overall stability of individual eating habits or a more situated tendency to repeat their first-time choices in the second session with exactly the same offerings. However, the multisensory condition was evaluated as very pleasant by the participants and positive emotional effects were reported based on the selection of emotion terms. In general, the feedback from participants regarding the whole experiment was positive; they valued the free, fresh and appealing buffet service and most were very satisfied with their meal.

Comparison with previous studies is challenging as we are not aware of studies using the same type of real-life but controlled lunch buffet settings in multisensory conditions. Previous studies used different study protocols, populations, buffet food selection and sensory primers. The buffet setting studies evaluated, for example, food choices of normal weight and overweight subjects [26]. Buffet meal intakes by different bitter taste sensitivity groups [27] or taste receptor genotype groups [28] were compared. In a multisensory study setting in Italy [29], consumers evaluated tomatoes and wild rocket in an immersive environment using countryside landscapes and sounds as well as natural herbs as olfactory cues. The liking scores were reported to be higher in the immersive environment compared with the traditional sensory laboratory setting [29]. In a self-service buffet setting, a priming experiment consisted of creating a leafy environment with green plants and an odor of herbs. The priming condition reduced the total energy intake [30]. Most previous studies have evaluated single foods or beverages in immersive conditions whereas this study provides new information about food consumption at a salad buffet in a multisensory environment.

Priming with food odor has been hypothesized to affect food selection but the results are controversial. Exposure to a fruity odor (pear) was found to increase the likelihood of selecting a fruity dessert [31]. Mors et al. [32] reported that priming with a bread or cucumber odor did not affect lunch choices but odor condition was associated with a self-reported positive mood. In the present study, exposure to the orange odor did not increase the selection or consumption of orange in the salad buffet; the trend seemed to be slightly the opposite. The auditory contribution to food perception was reviewed by Spence et al. [33]. Most previous studies have focused on the effect of music genre on the perception of a single flavor or food [34] but what kind of background sounds are most appropriate for lunch room conditions is not known. Different nature sounds, including birdsong, have been related to stress recovery and restorative benefits although restorative perceptions may vary between different bird species [35].

The buffet food selection in the present study was colorful and consumers were previously reported to value visually attractive and colorful salads [36]. External visual stimuli including colors of the food package, plates or cups may be associated with food perception [37]. Individuals differ in their associations of the color of liquid samples with taste qualities, pleasantness or healthiness [38]. In the present study, the tablecloth on the serving trolley was white while the color hue of the lighting and the color of the landscape were greenish in the multisensory condition. The color of the lighting may also have affected the color perception of food items offered in the salad buffet. Schifferstein et al. [39] reported that colored backgrounds affected the perceived attractiveness of vegetables but optimal background colors differed substantially for various vegetables. According to Hasenbeck et al. [40], yellow lighting increased the willingness to eat bell peppers. Because our buffet included food items with various colors, evaluating which colors of lighting would most effectively increase the attractiveness of vegetables and fruit was difficult. Complex landscape scenes present various colors and the effects of various pictures or scenes may be difficult to interpret. Investigating the effects of single sensory stimuli provides important information but in the multisensory context several aspects are combined. Real-life studies in restaurant settings also combine many sensory stimuli both in the food and in the environment. Therefore, the multisensory approach on consumer behavior and experience are challenging research topics.

Comparing study results focusing on emotions is difficult as the emotion terms vary and the results may be specific to the study population, study setting and the tested products. Few studies have related emotions specifically to the eating or meal situation [41,42]. In this study, the subjects selected the terms related to the whole meal situation and we do not know if their emotions were more related to the food eaten or the multisensory eating environment. We did not ask for opinions separately about different components (odor, lighting and sound) in the multisensory room. The reactions may be individual and some people may report unpleasant emotions associated with musical or pictorial stimuli [43]. Some consumers report adverse effects such as headaches related to fragrances [44]; thus, room odors should be used with caution. More research is needed on what kinds of sensory stimuli, as well as their combinations in different eating environments and with different consumer groups, can support pleasant eating experiences. Emotion questionnaires rely on self-reported subjective ratings of emotions and other measures would also be useful in food research settings. The review by Kaneko et al. [18] recommended combining various instruments, including physiological, behavioral and cognitive measures, for evaluating emotions evoked by food experiences.

The strength of this study was that real intake with real foods was studied and not just food pictures or fake food models [45,46]. In comparison with self-estimated portions of food intake in many nutrition studies, here the food intake was accurately weighed [47]. The same subjects attended two visits in a randomized order. Recording the sessions with a head-mounted eye-tracker overcame the need to set up external video cameras in the laboratory and we think this made the session monitoring feel less intrusive for the subjects. The possible limitation may be the short exposure time to the multisensory environment while taking the food. However, we wanted to create a situation resembling the normal food selection phase in a lunch restaurant setting and not to have the subjects wait before selecting the food. The subjects were seated alone whereas in real restaurants there may be many other external stimuli present and other customers. Only women were included in this study and men may have different preferences for lunch buffet foods as well as greater energy needs, requiring larger portions. The number of subjects was rather small but comparable with other buffet setting studies [26]. Only one subject was attending at a time and the preparation, serving and weighing of various fresh food items was rather laborious and time consuming. In future, real lunch restaurant buffets with a larger study population including both sexes could be studied.

In conclusion, the multisensory room conditions in this study did not change the food intake of the subjects. Fresh, colorful and a varied vegetable selection at lunch is appealing and could promote the consumption of vegetables and sustainable eating habits. In addition to fresh vegetables and fruit, salad buffets usually include other components and thus the overall nutrient composition and healthiness of the lunch depends on individual consumer choices. A pleasant and relaxing ambience may elicit positive feelings and thus enhance meal satisfaction and wellbeing [48]. The promotion of positive eating situations among various consumer groups deserves further study.

## Figures and Tables

**Figure 1 foods-09-01349-f001:**
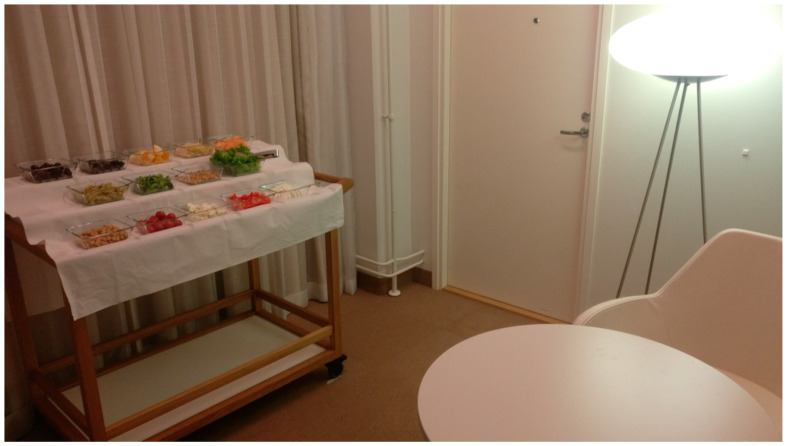
Serving trolley with the buffet foods.

**Figure 2 foods-09-01349-f002:**
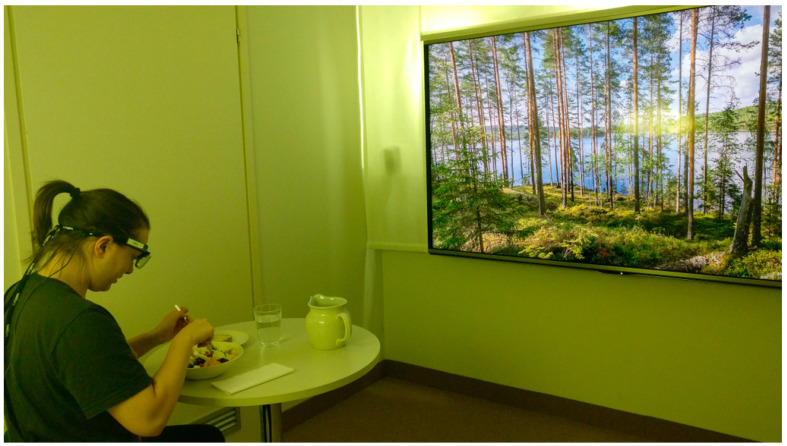
Lunch under a multisensory condition.

**Figure 3 foods-09-01349-f003:**
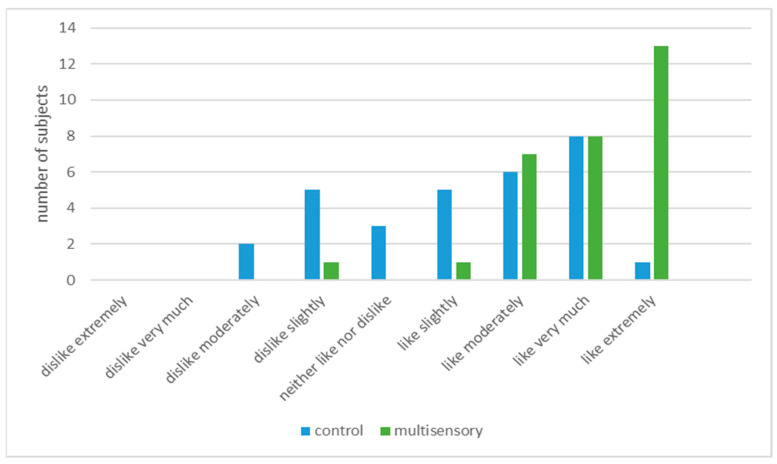
Liking of the test environment.

**Figure 4 foods-09-01349-f004:**
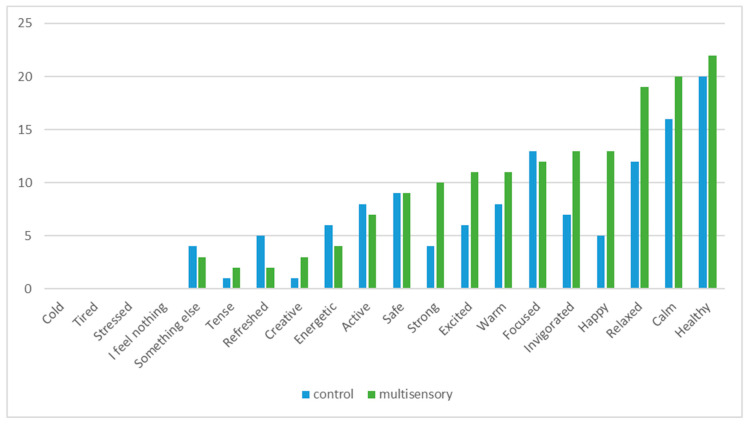
Selection of the emotion terms (n) in different conditions.

**Table 1 foods-09-01349-t001:** Foods served and serving size.

Food Color	Foods	Type, Preparation	Serving Size (g)
Black	Kalamata olive	canned, strained	150
	Black grape	rinsed	240
Green	Broccoli	frozen, defrosted	180
	Ice lettuce	rinsed, ripped to pieces	100
Red	Cherry tomato	rinsed	240
	Red bell pepper	rinsed, chopped	200
Beige	Chickpeas	canned, rinsed, strained	240
	Salted peanuts		140
Orange	Orange	peeled, cut	250
	Cantaloupe melon	peeled, cut	200
White	Mozzarella cheese	cut into slices	240
	Feta-type cheese	cubes, strained	210
Pasta	Pesto pasta	cooked pasta, cooled, mixed with pesto sauce 1:7	205
	Aioli pasta	cooked pasta, cooled, mixed with aioli mayonnaise 1:7	205

**Table 2 foods-09-01349-t002:** Foods consumed (mean, SD grams) at different conditions (control vs. multisensory).

Food	Food Intake Control (g) Mean (SD)	Food Intake Multisensory (g) Mean (SD)
Kalamata olive	14 (13)	14 (14)
Black grape	25 (18)	29 (16)
Broccoli	32 (21)	31 (22)
Ice lettuce	22 (13)	21 (14)
Cherry tomato	38 (24)	35 (22)
Red bell pepper	20 (17)	19 (17)
Chickpeas	17 (24)	15 (19)
Salted peanuts	7 (8)	7 (7)
Orange	38 (30)	33 (27)
Cantaloupe melon	43 (25)	38 (19)
Mozzarella cheese	36 (22)	32 (22)
Feta-type cheese	30 (20)	28 (18)
Pesto pasta	34 (27)	35 (29)
Aioli pasta	14 (17)	16 (25)
Total weight of the portion	372 (98)	354 (100)

The multisensory condition included a forest landscape on screen, birdsong and orange scent. *p*-values all non-significant (Wilcoxon signed-rank test).

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
