# Peer review of "Food Consumption and Emotions at a Salad Lunch Buffet in a Multisensory Environment"

_foods, 2020, doi:10.3390/foods9101349_

Round 1

Reviewer 1 Report

Authors improved the manuscript quite a lot. They explained the issues pretty well brought up by the reviewer.

However, I still concern some of the things.

  1. Discussion still contains too much information not relevant to the result of the study. There are not much result to show but authors tried to explain too long.  
  2. There were several items in the multisensory condition and they would have worked together. If the effect of any one of the items was to be shown, other items should have been controlled to be the same. Here only two conditions were compared; Controlled vs multisensory. The effect should be explained more carefully. Direct comparison shouldn't be made. 

Author Response

REVIEWER 1

Authors improved the manuscript quite a lot. They explained the issues pretty well brought up by the reviewer. However, I still concern some of the things.

Discussion still contains too much information not relevant to the result of the study. There are not much result to show but authors tried to explain too long.  There were several items in the multisensory condition and they would have worked together. If the effect of any one of the items was to be shown, other items should have been controlled to be the same. Here only two conditions were compared; Controlled vs multisensory. The effect should be explained more carefully. Direct comparison shouldn't be made.

OUR RESPONSE: We were grateful for this opportunity to improve our manuscript and we are happy that the reviewer valued our responses and revisions. We still want to point out that in food research including real consumption part, the multisensory research environments are still rather new approach and combinations of many sensory stimuli at the same time (odor, sound, visul stimuli) have seldom been investigated.

We fully agree that in order to understand the sensory stimuli better, those should be studied separately first. In fact, in our multisensory lab, we have systematically investigated ambient sensory stimuli alone or their various combinations in separate other study settings. Those sessions did not include food intake and in this study we wanted to have food choice and intake measures under multisensory environment. Many food intake sessions with various sensory stimuli combinations would be very laborious for the researchers and the subjects as well. In fact, the real-life studies in real restaurant setting also always combine many sensory stimuli, both in the food and the environment, and it seems to be impossible to separate all the effects. We have now added a sentence also to the discussion section, page 9, highlighting this important research challenge.

In the results section we present the results of comparison control vs. multisensory for the food intake (Table 2), liking of the test environment (Figure 3) as well as emotion terms (Figure 4). All of these aspects are also dealt with in the discussion section. We are aware that we cannot know what separate items in the sessions were related for example to the perceived emotional states. However, multisensory condition seemed to be related to more positive ratings. As we have pointed out, more research with more variable study setting and consumer groups is needed.

Reviewer 2 Report

Dear Editor,

Thank you for the opportunity to evaluate Food consumption and emotions at salad lunch 3 buffet in multisensory environment for Foods. Overall the study is clearly written and addresses a both relevant and interesting topic, using an intelligent design. 

Nevertheless some concerns hamper the potential merit of the study and might need some additional attention. 

Introduction:

Clearly a relevant and at present a popular topic to be studied. Since some of the issues addressed have been investigated extensively recently, the authors might benefit from (some of) the work of Daisuke Kaneko and professor Alexander Toet to strengthen their proposition.

Methods:

Line 119, please provide an appropriate reference at "In a previous study... "

Line 119/120, On what criteria were these two ambiances / conditions selected?

 Line 132, "Orange scent": Multisensory coherence? Might be of relevance to present a convincing / gestalt experience. Orange not necessarily 'matches' pines woods? Especially in relation to the appreciation to 'taste' (if food is congruent with ambiance, appreciation might significantly increase?)?

Line 148, "emotion questionnaires": Please provide more information on the relevant questionnaires here (e.g. titles, abbreviations and scoring formats, with the already provided references). 

Line 178, using the eye tracking technology in this way seems to be a rather complicated method to obtain such data. Even a bit 'inconvenient' for the subjects / participants? 

Line 181, "Analyses reported in another article": The authors regularly refer to 'another article' in relation to potentially relevant data. It does suggest piecemeal publication of data that could (or should?) be combined, e.g. in this paper? 

Line 189, What features of the data required a "non-parametric" methodological strategy? 

Results: 

Line 205 / 206, If "non-significant" please do not refer to potential differences. 

Line 225 / 226, With regards to the assessment of the multisensory condition: Was it checked for potential sequencing effects?: there may be significant higher appreciation for the multisensory environment when it was presented in the second visit (as compared to the first) ("Hey!, this is much more comfortable than last week!")?

Conclusions:

Line 273, "... slightly the opposite": Potentially due to incongruence with other sensory modalities? (see previous comment) What may indeed cause a negative affective assessment.  

Line 295 / 296, Kaneko et al., 2018 presented a rather comprehensive overview of methods to assess emotional responses to food experiences what might provide some alternatives to consider (in the discussion). 

Author Response

REVIEWER 2

Thank you for the opportunity to evaluate Food consumption and emotions at salad lunch buffet in multisensory environment for Foods. Overall the study is clearly written and addresses a both relevant and interesting topic, using an intelligent design.
Nevertheless, some concerns hamper the potential merit of the study and might need some additional attention.

Introduction:
Clearly a relevant and at present a popular topic to be studied. Since some of the issues addressed have been investigated extensively recently, the authors might benefit from (some of) the work of Daisuke Kaneko and professor Alexander Toet to strengthen their proposition.

OUR RESPONSE: Thank you for pointing out these very interesting and topical publications by Kaneko and Toet. Their excellent review (Kaneko et al. 2018) about methods for evaluating emotions is very useful for the researchers in food science. We have now included this review to the introduction section and further in the discussion. Also another paper by Toet et al. 2018 (EmojiGrid) is now mentioned.

Methods:

Line 119, please provide an appropriate reference at "In a previous study... "

RE: This study part has not been reported or published yet and thus we have rephrased the sentence to refer to our pilot-test phase, which was the basis for the selection of the multisensory condition (see also answer below)

Line 119/120, On what criteria were these two ambiances / conditions selected?

RE: The other condition was plain control and conditions for the multisensory condition were selected after pilot testing. Various sensory stimuli in the multisensory lab, different landscape scenes with different lightning conditions, birdsong soundtracks and ambient odors/scents, were systematically pilot-tested and feedback from the pilot subjects was collected. After the pilot-testing the researchers selected the combination of the conditions for this study part.

Line 132, "Orange scent": Multisensory coherence? Might be of relevance to present a convincing / gestalt experience. Orange not necessarily 'matches' pines woods? Especially in relation to the appreciation to 'taste' (if food is congruent with ambiance, appreciation might significantly increase?)?

RE: We thank the reviewer for pointing out this important issue about the selection of odor, should it be matching with the visual/landscape or with the food offered. In the pilot-test phase we also had a pinewood/forest like odor to test, but some pilot subjects reported that it would not match with the food. The orange scent was chosen because slices of orange were offered in the salad buffet (possible priming effect for orange intake) and it was generally regarded as fruity odor that could match the fruit and vegetable selection of the buffet. Moreover, because of instrumental reasons we had limited options to make a selection of scents that were suitable for the device we used.

Line 148, "emotion questionnaires": Please provide more information on the relevant questionnaires here (e.g. titles, abbreviations and scoring formats, with the already provided references).

RE: There are various different emotion term questionnaires published and used in relation to food studies recently, and we screened for many questionnaires in the planning phase of the study. In our methods paragraph, these two emotion questionnaires mentioned in line 148 formed the basis for the term testing and selection for the Finnish consumers. These were modified and only part of the original emotion terms were selected, and thus the original scoring format was not used.

Line 178, using the eye tracking technology in this way seems to be a rather complicated method to obtain such data. Even a bit 'inconvenient' for the subjects / participants?

RE: As mentioned in the methods section, two sets of data were collected but analysed and reported separately (food intake and emotions reported here as well as detailed eye-tracking results reported elsewhere). The time used for eating was easily obtained from the eye-tracking recording and therefore only this result was used here. Naturally, if we had only wanted to measure the time used for eating it would have been possible much easily with a simple timer. The comments from the subjects were also positive, they did not feel the eye-tracking device as inconvenient but they were excited about this new method.

Line 181, "Analyses reported in another article": The authors regularly refer to 'another article' in relation to potentially relevant data. It does suggest piecemeal publication of data that could (or should?) be combined, e.g. in this paper?

RE: As mentioned in the methods sections, two sets of data were collected but analysed and reported separately. This manuscript focuses on food intake and emotions. The eye-tracking part included in the session focused on the visual attention toward food items (eye movement variables). This another article has separate aim, specific eye-tracking methodology and results to report (first author eye-tracking expert Marjaana Puurtinen). The eye-tracking part has a lot of new type of methods and data to report and it is impossible to combine that in this paper. For the subject it was part of the same study visit and therefore we wanted to present also the overview of the study procedure here.

Line 189, What features of the data required a "non-parametric" methodological strategy?

RE: The food intake measures were not normally distributed but rather skewed. Some subjects did not take some foods at all while others had a very large amount of single food, depending on their personal preferences.

Results:

Line 205 / 206, If "non-significant" please do not refer to potential differences.

RE: Thank you for this comment, we have now removed this sentence.

Line 225 / 226, With regards to the assessment of the multisensory condition: Was it checked for potential sequencing effects?: there may be significant higher appreciation for the multisensory environment when it was presented in the second visit (as compared to the first) ("Hey!, this is much more comfortable than last week!")?

RE: We also checked this effect but there was no significant difference depending on the visit order. The control and multisensory conditions were randomised for each participant. These subjects had also visited in the multisensory lab under different conditions before, and in general, they seemed to feel comfortable in the room.

Conclusions:

Line 273, "... slightly the opposite": Potentially due to incongruence with other sensory modalities? (see previous comment) What may indeed cause a negative affective assessment. 

RE: We have thought about this possible negative trend but it is difficult to speculate what could be the cause. We did not ask separately about the perception or liking of the orange scent in this study. However, individual odor perception has previously shown to vary a lot and perhaps some subjects did not like citrus odors. It is also possible that the odor was perceived incongruent with the pine forest landscape. Other studies have also found that priming with specific food odors did not affect food intake while had some association with positive mood (e.g. Mors et al. 2018). Therefore, much more research is needed about the food odors or other ambient odors in meal situations.

Line 295 / 296, Kaneko et al., 2018 presented a rather comprehensive overview of methods to assess emotional responses to food experiences what might provide some alternatives to consider (in the discussion).

RE:  The review by Kaneko et al. is very interesting and we have now mentioned it already in the introduction and again in the discussion. The self-reported subjective ratings with emotion questionnaires would be important to complement with other methods, as suggested by Kaneko et al. We have now added this important consideration also in the discussion section, page 9.

This manuscript is a resubmission of an earlier submission. The following is a list of the peer review reports and author responses from that submission.

Round 1

Reviewer 1 Report

The main weakness is the rather small sample. In sensory research, it is common to have at least around 80-100 consumers in a sensory test (Meiselman, 2013). A study by Delarue et al. (2019), comparing the lab and immersive settings, worked for instance with 249 participants. Further, the introduction is really short, and should elaborate more on other studies comparing the effect of lab vs immersive & emotions.

Introduction

Introduction is rather short, some more information about previous studies comparing the impact of lab vs. immersive is recommended. Further, the link regarding the emotions should be elaborated. Why do the authors expect that this might have an influence? (any prior research? Theoretical underpinning?)

L51-55: relevance for this study? => this may be omitted, better to focus on the effect of the lab & the emotions

Methods

L67: please mention this ‘another article’

L142 The term selection was pretested with 142 Finnish consumers in a previous study setting focusing on ambient odors in the multisensory room. => so, the this pretest did not involve food consumption? How valid is then this pretest?

L145: lunch at 10.45 am sounds as rather early. Why was chosen for this time and is this a normal lunch time in Finland, especially given that the other session was at 12.30 am.

Reviewer 2 Report

This manuscript contains some interesting emotion related study. However, the purpose of the study is not clear and this MS delivers too little new information. And the design of the study has some  problems.

  1. The methods used seems to have weakness to fulfill the object of the study. And the result and discussion were not logically or systemically described.
  • 30 female subjects are too few for the consumer study.
  • The subjects participated in both sessions and this might have blurred the results of the study. They could have noticed the intention of the researchers.
  • Multisensory environment was compared with control condition and each environment was compared with control in the result and discussion. This might have misled the result.
  • The subjects took as much salad as they wanted and the trolley was taken away. They could have felt like having more after they finished the salad they took.
  1. Details

-line 63; subjects were the one who previously participated in extensive sensory tests and further taste testing in a multisensory laboratory. I wonder if they were the proper subjects for this type of the study. General consumers and bigger number of them would have been proper subjects.

-line 109; there should be more description for the multisensory lab. What was the basis for choice of the conditions?

-line 133; Why were the numbers of category in the scales different? Some were 4 and the others, 9. There was questionnaire on liking of the test environment but not on the liking of the food. Any reason? Reference 17 was on emotion questionnaires but not ref 18 (well sense profile). Who selected the emotion terms and how? It is indicated that the terms were tested with consumers but actual testing was done by experienced subjects (if not trained) What was the logic behind? Why was CATA used? There were only two conditions (control and multisensory conditions) and I wonder if the authors thought using CATA could differentiate two conditions. Yes, it could. But it could have shown better if rating was also used as in RATA.    

-Times of the session is rather peculiar. 10:45am or 12:30pm. What was the principle for choosing these times? It should be explained.

-line 175; the main focus of the study was on food intake and emotional measures according to the condition of the test room. However, food intake measure could have been biased because the salad trolley was taken away after the subject put the salad as much as they want on the plate. They could have more if the trolley was left in the room, or they could have left some of the salad. Since they were required to take the amount as much as they like, so they could have felt that they have to finish it. It seems that the amount was fixed soon after they entered the room without being affected much by the environment.

-It would be better, if there are subtitles in Results section.

-In Table 2, food intake doesn’t seem to be very different, however, out of 14 items of salad, amount of intake in 9 items were higher in the control condition and 3, in the multisensory condition with two ties. It needs to be explained.

-I feel that the emotion results would have been the same as without consuming salad even though the multisensory condition is very complex and salad items were many. These all make the situation of the study not clear thus making the purpose of the study vague.

-Discussion includes too much information from previous studies not relevant to the current study. It should be for the interpretation of the results of the study. Much of the information, here should go to the introduction to justify the study.

-line 264; Since multisensory effects worked together, discussion of individual effect could mislead the results.

-line 305; Authors indicated the strength of the study was that real intake with real foods was studied ~~. Here, the procedure of intake measure could have made with bias as mentioned above. Very complex condition was involved in the measurement of very simple matters. This could mislead the results and improper discussion and conclusion.

-Authors indicated future study is needed in several parts in this MS, however, some of them should be included for this study to have more meaningful results.